# Resistance and Aerobic Training Were Effective in Activating Different Markers of the Browning Process in Obesity

**DOI:** 10.3390/ijms25010275

**Published:** 2023-12-24

**Authors:** Lidia Passinho Paz Pontes, Fernanda Cristina Alves Nakakura, Nelson Inácio Pinto Neto, Valter Tadeu Boldarine, Paloma Korehisa Maza, Paloma Freire Santos, Felipe Avila, Artur Francisco Silva-Neto, Hanna Karen Moreira Antunes, Ana Raimunda Dâmaso, Lila Missae Oyama

**Affiliations:** 1Department of Physiology, Universidade Federal de São Paulo, Escola Paulista de Medicina, São Paulo 04023060, Brazil; lidiapontes.ap@gmail.com (L.P.P.P.); fernanda3496@hotmail.com (F.C.A.N.); nelson.inacio@unifesp.br (N.I.P.N.); valter.tadeu@unifesp.br (V.T.B.); pkmaza@unifesp.br (P.K.M.); paloma.freire92@gmail.com (P.F.S.); f.avila@unifesp.br (F.A.); artur.francisco@unifesp.br (A.F.S.-N.); ana.damaso@unifesp.br (A.R.D.); 2Department of Psychobiology, Universidade Federal de São Paulo, Escola Paulista de Medicina, São Paulo 04023060, Brazil; hanna.karen@unifesp.br

**Keywords:** obesity, physical training, browning, adipose tissue, nutrition, FGF21, PGC1α, PPARγ, UCP1, irisin

## Abstract

Lifestyle changes regarding diet composition and exercise training have been widely used as a non-pharmacological clinical strategy in the treatment of obesity, a complex and difficult-to-control disease. Taking the potential of exercise in the browning process and in increasing thermogenesis into account, the aim of this paper was to evaluate the effect of resistance, aerobic, and combination training on markers of browning of white adipose tissue from rats with obesity who were switched to a balanced diet with normal calorie intake. Different types of training groups promote a reduction in the adipose tissue and delta mass compared to the sedentary high-fat diet group (HS). Interestingly, irisin in adipose tissues was higher in the resistance exercise (RE) and aerobic exercise (AE) groups compared to control groups. Moreover, in adipose tissue, the fibroblast growth factor 21 (FGF21), coactivator 1 α (PGC1α), and peroxisome proliferator-activated receptor gamma (PPARγ) were higher in response to resistance training RE compared with the control groups, respectively. Additionally, uncoupling protein 1 (UCP1) showed higher levels in response to group AE compared to the HS group. In conclusion, the browning process in white adipose tissue responds differently toward different training exercise protocols, with resistance and aerobic training efficient in activating different biomarkers of the browning process, upregulating irisin, FGF21, PGC1α, PPARγ, and UCP1 in WAT, which together may suggest an improvement in the thermogenic process in the adipose tissue. Considering the experimental conditions of the present investigation, we suggest future research to pave new avenues to be applied in clinical practices to combat obesity.

## 1. Introduction

Obesity is a multifactorial disease because of many factors, including genetics and poor lifestyle, and low quality of diet and sedentarism may increase its prevalence worldwide. Nowadays, its control is one of the world’s greatest challenges, since its prevalence has reached the state of a global pandemic [1]. This disease is a public health problem affecting not only the whole health of individuals but also the economy of many countries [2].

On the other hand, physical exercise increases the resting metabolic rate and thermogenesis, improves insulin sensitivity, and controls dyslipidemia and cardiometabolic risks. Corgosinho et al. [3] reinforce the value of interventions in lifestyle, including physical exercise and nutritional adjustment, to control the inflammation present in obesity. Furthermore, Hsu et al. [4] showed that aerobic training reduces both body and fat mass indexes, resistance training improves both body composition and physical performance, and a combination of aerobic and resistance training improves walking speed and reduces fat mass.

Interestingly, adipose tissue (AT) has been considered a metabolically active and highly heterogeneous endocrine organ [5] that is related to the regulation of glucose and lipid metabolism, insulin sensitivity, inflammatory response, and thermogenesis [6]. Classically, AT has been divided into white adipose tissue (WAT) and brown adipose tissue (BAT). This organ comprises several well-defined deposits mainly located at the following two body levels: superficial (subcutaneous deposits) and deep (visceral deposits) [7]. WAT stores excess energy as triglycerides, acts as a thermal insulator, protects internal organs against mechanical damage, and secretes adipokines, which are related to inflammation, angiogenesis, and metabolism. On the other hand, BAT has thermogenic potential, helps maintain an adequate body temperature, and may have an important role in weight loss [7].

Notably, a significant phenomenon has been described in the literature in recent decades: WAT browning. In this process, depending on the stimuli, white adipocytes transdifferentiate into brown ones, which are called beige adipocytes and have a high thermogenic capability, in addition to contributing to energy expenditure [8]. Stimuli that may increase thermogenesis include exposure to cold, diet, response to physical exercise, and drug administration [8,9,10]. This evidence has opened up new therapeutic possibilities in the treatment of obesity and related disorders [8,9].

Fedewa et al. [11] showed that different forms of physical training promoted a reduction in body mass and adiposity. Additionally, several studies have demonstrated that physical training can modulate the expression of thermogenic genes in WAT, promoting an increase in the body’s energy expenditure. It has also been observed that associating healthy nutrition with physical exercise has potentially beneficial effects on the control of obesity and its comorbidities, mediated by AT remodeling [12]. Furthermore, Calcagno et al. [13] reported on the thermal effect of food as a potentially modifiable component of energy expenditure and the relationship between balanced diets and physical activity that stimulate this effect.

The significant markers expressed during WAT browning are uncoupling protein 1 (UCP1), considered a key regulator of thermogenesis; peroxisome proliferator-activated receptor gamma (PPARγ), which plays important roles in glucose and lipid metabolism [14]; and its coactivator 1 α (PGC1α), which drives the development of the thermogenesis [15]. Moreover, the increase in irisin caused by exercise induces the browning process, and fibroblast growth factor 21 (FGF21) plays a significant physiological role in WAT thermogenesis [16].

In addition, studies evaluating the effects of physical training on the adaptive responses of human and rodent WAT and BAT show contradictory data, which makes this investigation field relevant [17]. Few studies have compared different types of training with changes in diet and associated these with activators that could influence the WAT browning markers as a potential therapeutic target in treating obesity and its associated disorders. Therefore, the objective of the present study was to investigate rats with obesity induced by a high-fat diet that, halfway through the experiment, were switched to a balanced normal calorie diet and also underwent resistance, aerobic, and combination physical training, to evaluate the effect of different protocols of training on the markers involved in the WAT browning process.

## 2. Results

### 2.1. Effect of the High-Calorie, High-Fat Diet on the Induction of Obesity

It was possible to observe that 16 weeks of ingesting a high-calorie, high-fat diet were effective in inducing obesity in the sedentary high-fat diet group (HS). In the body parameters observed in all experimental groups (Figure 1), we observed that adiposity, body weight, and total amount of fat in the carcass were significantly less in the groups that underwent training than in the HS group. The total food ingestion (Figure 1D) was significantly greater in the groups aerobic exercise (AE) and combination exercise (CE) than in the HS group. The biochemical and body parameter data demonstrate the effectiveness of this study’s obesity model (more detailed data are described in the Appendix A).

### 2.2. Progression of the Load and Performance in the Training Groups

After eight weeks of training in aerobic (AE), resistance (RE), and combination (CE) groups, it was possible to observe the progression of aerobic performance in AE and CE groups and the progression of the loads in RE and CE groups. 

According to the final strength test (Figure 2A), the group combination exercise (CE) showed a significantly smaller lifted weight when compared to the group resistance exercise (RE) (*p* < 0.001). As for the load progression in the resistance training (Figure 2B), both RE and CE groups showed a significant increase throughout the experimental protocol (*p* < 0.001). Nevertheless, we observed that group RE had a significantly greater increase than group CE in all eight weeks of training (*p* < 0.001). These findings indicate that this comparatively greater load increase during the training results from the specificity of the strength training.

Regarding the progression of aerobic performance according to velocity (m/min) during the eight weeks of training (Figure 2C), it was observed that group AE had a significant increase in the fourth and eighth weeks when compared to the baseline (*p* < 0.01), group CE showed greater speed in the eighth week when compared to the baseline, and group AE showed greater speed (m/min) when compared to group CE in the fourth week of training (*p* < 0.01). As for the aerobic performance according to distance (m) (Figure 2D), it could be observed that the distance covered by group AE was longer relative to the baseline in the fourth and eighth weeks of training (*p* < 0.01), the distance was longer for the baseline for group CE in the eighth week (*p* < 0.01), and group CE showed a shorter distance covered in comparison with group AE in the fourth and eighth weeks of training (*p* < 0.01).

### 2.3. Adipose Tissue Browning Mediated by Physical Training

Irisin concentrations (Figure 3) in retroperitoneal AT were significantly greater in group RE when compared to group HS (*p* < 0.02). In mesenteric AT, group AE presented significantly greater concentrations in comparison with group CS (*p* < 0.02); in subcutaneous AT, group RE showed greater concentrations than control sedentary group CS (*p* < 0.01).

FGF21 concentrations (Figure 4) in retroperitoneal AT were significantly greater in the resistance training group in comparison with groups CS (*p* < 0.01) and HS (*p* < 0.02). In subcutaneous AT, the resistance training group showed significantly greater FGF21 concentrations than in groups CS (*p* < 0.04) and HS (*p* < 0.03), while in mesenteric AT no significant difference in FGF21 concentration was observed between the groups.

The quantification of the proteins involved in AT browning, obtained by Western blotting of mesenteric AT, is shown in Figure 5. A significant increase in PGC1α concentration was observed in group RE when compared with group HS (*p* < 0.02). Group RE also showed a significantly increased PPARγ concentration about group HS (*p* < 0.04), while UCP1 concentration was found to be significantly elevated in group AE in comparison with group HS (*p* < 0.01).

In retroperitoneal AT, no significant differences were observed in the protein concentrations of PGC1α, PPARγ, and UCP1 between the experimental groups (described in Appendix A).

In subcutaneous AT (Figure 6), a significantly increased PGC1α concentration was observed in group RE when compared to groups CS (*p* < 0.04) and HS (*p* < 0.01). No significant differences were observed in PPARγ and UCP1 concentrations between the groups.

## 3. Discussion

Interestingly, the present study demonstrated that aerobic and resistance training in isolation showed more significant responses than combination training in the same training session. It could be observed that both aerobic and resistance training could modulate irisin, FGF21, PGC1α, PPARγ, and UCP1 in WAT, which together may suggest an improvement in the thermogenic process in the adipose tissue.

Additionally, we were able to show that irisin concentrations were modulated in a dependent manner, considering the different depots of white adipose tissue analyzed. Another study conducted with Wistar rats demonstrated that eight weeks of continuous and interval aerobic training resulted in a significant increase in irisin concentration in visceral AT, showing the effectiveness of this type of training in modulating irisin, which is related to risk factors for metabolic syndrome [18]. However, Rocha-Rodrigues et al. [19] demonstrated that aerobic training for eight weeks did not change irisin concentration in Sprague Dawley rats with obesity induced by a high-fat diet. Moreover, a study in humans with excess weight and obesity conducted by Kim et al. [20] demonstrated that after eight weeks of physical training, circulating irisin was higher in the resistance training group, whereas the aerobic training group did not show any changes in this biomarker.

Abdi et al. [21] observed that combination training in an eight-week protocol in Wistar rats can play a role in mitigating the metabolic disorders caused by obesity and type-2 diabetes, but irisin concentrations in AT showed a mere tendency to increase in response to the training. The literature shows evidence that irisin can reduce diet-induced obesity and increase the number of cells similar to brown adipocytes, and that the expression of brown-adipocyte-specific markers in WAT stimulates thermogenesis and increases energy expenditure [22]. The findings of the present study suggest that irisin appears to be a mediator capable of promoting positive actions that are stimulated by resistance and aerobic training and related to their effects on WAT metabolism, and this can be a potential therapeutic target for treating obesity.

Another important result observed in the present investigation is that resistance training promotes an increase in FGF21 in both visceral and subcutaneous adipose tissues. Importantly, FGF21 is a key regulator of the differentiation of WAT into BAT, resulting in greater energy expenditure and thermogenesis [23]. Corroborating this, Geng et al. [24] demonstrated that physical training conducted on a treadmill for four weeks improved the metabolic dysfunction induced by obesity through an increased sensitivity to FGF21 in the white adipose tissue of male obese C57BL/6J mice. However, Morville et al. [25] demonstrated divergent effects of resistance training on FGF21 concentration in humans: the group undergoing aerobic training group presented higher FGF21 concentrations than the group undergoing resistance training.

Motahari et al. [26] demonstrated that combination training did not change FGF21 concentration in adults after 12 weeks of training. The results of the present study are in line with the literature that proposes FGF21 as an important mediator for the metabolic benefits induced by exercise in rodents. Considering these findings, the present study demonstrated that FGF21 induction appears to act on positive metabolic effects promoted by either resistance or aerobic training alone. In contrast, combination training did not result in these changes in white adipose tissue. Future research is suggested to identify probably mechanisms related.

Morton et al. [27] conducted a study with C57BL/6 mice with access to running wheels for six experimental weeks. That study showed that mice fed with a high-fat diet that practiced physical exercise presented increased UCP1 and PGC1α levels when compared with the control group. However, Aldiss et al. [28] applied a four-week swimming training protocol to obese Sprague Dawley rats in thermal neutrality conditions and demonstrated that physical exercise did not induce any increase in UCP1 and PGC1α levels in AT under these conditions. These controversies may suggest that different lineages of rats, different types of exercise conditions, and the duration of the exercise protocol may differently affect the obtained results in some biomarkers of the browning process. These findings need to be confirmed in both experimental and clinical conditions in view to determine its applicability in humans. 

Furthermore, Vosselman et al. [29] assessed the markers involved in the browning of subcutaneous adipose tissue in men after chronic aerobic training. The markers were measured after exposure to light cold, and no increased UCP1 concentrations were detected. Additionally, no significant difference in PGC1α levels was observed between trained and sedentary groups. However, Li et al. [30] conducted an eight-week study in obese rats, and increased PPARγ concentrations were observed both in adipose tissue and in plasma in all trained groups when compared to the control group.

Picoli et al. [31] observed that both resistance and aerobic training for eight weeks induced browning in subcutaneous and visceral adipose tissue in Swiss mice, showing a significant increase in UCP1 and PGC1α concentrations in AT. Ziegler et al. [32], in turn, conducted a study with mice that underwent resistance and aerobic training for 10 weeks and demonstrated that PGC1α concentrations were greater in the aerobic training group than in the resistance training group, while no effect of exercise on UCP1 concentrations was observed. Stotzer et al. [33] showed that 10 weeks of training on stairs with progressive loads in ovariectomized Sprague Dawley rats increased PPARγ concentration.

Norheim et al. [34] studied combination training for two weeks in overweight humans. In that study, it was not possible to observe any increase in UCP1 and PGC1α concentrations in subcutaneous AT. Similar findings were reported in the study by Stinkens et al. [35], in which 12 weeks of combined training did not change PCG1α and UCP1 concentrations in subcutaneous AT in obese humans. Moreover, the effect of the morphology of adipose tissue in situations of combination training remains ambiguous [22], and few studies have assessed the effect of combination training on the parameters studied in the present research. Therefore, further investigations are needed to evaluate the effect of combination training in different protocols on browning markers.

The metabolic benefits of physical training induce key modulations in adipose tissue metabolism. In addition, the literature shows that divergent types of training present different effects on WAT browning [36]. Contradictions can be found in the literature regarding the response to each type of training, but the effectiveness of the browning mediated by physical training appears to be related to the type of training and the duration and intensity of each training type. 

It is important to point out the role of the sympathetic nervous system in this context. As reported by Dong et al. [37], one of the mechanisms by which exercise can regulate the browning process in white adipose tissue is by sympathetic nervous system activation. Murano et al. [38] showed a strong correlation between sympathetic nervous system activation and browning in different white adipose tissue depots in mice, and Efremova et al. [39] observed a similar correlation in humans. To emphasize the importance of the sympathetic nervous system on the browning process, Jimenez et al. [40] demonstrated that β-3 adrenoceptor knockout in mice impairs the browning process in the white adipose depot.

Although the combination training did not present any affection on browning markers, there is some tendency to increase irisin and FGF21 content in the subcutaneous adipose tissue. Likely, increasing the treatment length can effectively promote some alterations. Furthermore, as a limitation, gene expression analyses that might help explain some results were not performed. Furthermore, experimental studies cannot be linearly extrapolated to clinical practice. Although this is a limiting factor, such studies can still provide guidance and theoretical bases for the development of relevant approaches to combat obesity and metabolic disorders. Future studies are needed to expand the study of the pathways involved, examining each browning mechanism in different adipose tissue depots.

Finally, it is important to mention that independent types of physical training protocol in lifestyle changes can promote a reduction in visceral adiposity, in the variation of body mass, and in the total fat amount in the carcass when compared to the sedentary group that was fed a high-fat diet. The three groups that underwent physical training returned to the same feeding regimen as the control group during the training period. Indeed, Verheggen et al. [41] conducted a systematic review and meta-analysis and demonstrated that both physical training and diet decrease visceral adiposity in humans. Although the diet promotes a greater loss of body mass, training tends to have superior effects in reducing visceral adipose tissue. In agreement with the results of the present study, Zhang et al. [42] demonstrated that both body mass and visceral fat were significantly reduced after an eight-week swimming training with a weight load in obese rats. 

## 4. Material and Methods

### 4.1. Animals, Diet, and Experiment Groups

Fifty-six male Wistar rats (age: 2 months) originating from the Center for the Development of Experimental Models in Medicine and Biology (CEDEME) of the Federal University of São Paulo (UNIFESP), São Paulo City Campus, Brazil, and intended for the vivarium of the University’s Psychobiology Department were used. The whole experimental protocol was approved by the Animal Research Ethics Committee (process number: 9256270618). The animals were kept in a 12 h/12 h light-dark cycle in an environment with temperature and humidity controlled daily (22 °C and 35%, respectively). The animals were weighed, randomly assigned to one of 5 groups, and distributed into cages (n = 2 per cage). The experimental period was 16 weeks long, with the first 8 weeks used to induce obesity through a high-fat diet and the last 8 weeks used for treatment with physical training and switching the diet to the control one.

The groups were named as follows: sedentary group receiving a control diet (CS, n = 16); sedentary group receiving a high-fat diet (HS, n = 16); aerobic exercise group receiving a high-fat diet (AE, n = 8); resistance exercise group receiving a high-fat diet (RE, n = 8); and combination exercise group receiving a high-fat diet (CE, n = 8). Group CS was fed with a standard Nuvilab diet (2.87 kcal/g, 15% of total calories from fat; Nuvital Nutrientes S.A., Colombo, State of Paraná, Brazil), while groups HS, AE, RE, and CE received a high-fat diet (3.60 kcal/g, 45% fat—Nuvilab diet and lard added according to the proportions described in Appendix A).

Groups AE, RE, and CE received a high-fat diet for 8 weeks; these groups then received the standard Nuvilab diet for the next 8 weeks. The change in the diet aimed to return the animals to a normal pattern of obesity control through different types of physical exercise associated with a balanced diet with a normal calorie intake. Group HS received the high-fat diet, and group CS received the standard diet for the entire duration of the experiment. To assess the obesogenic effectiveness of the diet and validation of metabolic changes, some animals from groups CS (n = 8) and HS (n = 8) were euthanized after 8 weeks of obesity induction to check their biochemical and metabolic parameters during that period. Previous studies by our group have demonstrated the metabolic effects of the high-fat diet model [43,44]. The other animals from groups CS (n = 8) and HS (n = 8) were not euthanized until the end of the experiment, to check the same parameters and compare them at the end (described in Appendix A). The description of the experimental design can be found in Appendix A.

### 4.2. Body Mass Gain and Feed Consumption

The animals were weighed once a week to measure the gain in body mass. This was determined through the progression of body mass and its gain (Δ*M*) in all animals throughout the experiment. The body mass gain was analyzed through the difference between each animal’s mass at the start (*M_i_*) and at the end (*M_f_*) of the experiment (Δ*M* = *M_f_* – *M_i_*).

The ingestion of feed was checked 3 times a week throughout the experiment. The leftover feed was weighed once a week. The ingestion was measured as the difference between the mass of feed supplied and that of the leftovers.

### 4.3. Training Protocol

#### 4.3.1. Test Description and Strength Training

A strength test was performed at the beginning of each week, according to the method by Mônico-Neto et al. [45]. The animals used the training apparatus described therein, which allowed them to perform repetitions to move from the bottom to the top of a set of stairs until failure. The overload was progressively increased with 50% of the maximum load in the first set, 75% in the second, 90% in the third, and 100% in the fourth set, with 60 s intervals between sets. In the following climbs, 30 g were added in each attempt until failure [45].

The training consisted of 4 to 8 series of stair climbing, with progressively greater loads, and intervals of 60 s between series. These were performed 5 times per week, from Monday to Friday. The training apparatus allows the rats to perform 8–12 repetitions to move from the beginning to the top of the stairs. The overload in the first set was progressively increased according to the description of the strength test with ascents to failure [45]. Since this is a high-intensity protocol, a prophylactic rest day (on Wednesdays) was introduced from week 6 to improve performance. The rats performed 4–8 sets over the protocol weeks; the large interval can be justified by the exercise principle known as volume/intensity interdependence. This principle is related to exercise overload, which implies a performance improvement through volume and intensity. At the start of the training, the animals carried lighter loads (lower intensity) but performed more series (higher volume). This ratio was inverted over the weeks, and the rats were able to carry heavier loads fewer times. The resistance training is described in the Appendix A. However, it can be surely stated that this method is adequate for animal welfare, as no additional stress (such as electrical stimulation or food restriction) is needed.

#### 4.3.2. Effort Test and Aerobic Training

An adaptation process was conducted with the rats for 10 min over 3 consecutive days, with a speed in the 5–10 m/min range. The training for the aerobic effort test was conducted on an AVS^®^ treadmill. The animals underwent a physical exertion tolerance test with an initial speed of 5 m/min, increased every 3 min until the animal could no longer keep up with the running level.

The training was conducted for 8 weeks. In the 1st week, the rats were trained with 30% of the maximum speed (V_max_) achieved in the tolerance test for 30 min. In the following 3 weeks, the duration was increased to 40, 50, and 60 min, respectively, with an intensity of 30% of V_max_. From the 5th week, the duration was maintained, while exercise intensity increased successively to 40%, 45%, 50%, and 55% of V_max_. At the end of the 4th and 8th weeks of training, full V_max_ was used to adjust the exercise intensity. The aerobic training protocol is described in Appendix A.

#### 4.3.3. Combination Training

The combination exercise group underwent combined combination training with both aerobic and resistance exercises which were performed on the same day, beginning with resistance and followed by aerobic exercise. For the aerobic exercise, the rats were trained with 30% of the V_max_ achieved in a 15-minute tolerance test. In the following 3 weeks, duration was increased to 20, 25, and 30 min, respectively, with a 30% V_max_ intensity. Starting from the 5th week, a 30-min duration was maintained, but the intensity was successively increased to 40%, 45%, 50%, and 55% V_max_. The 1st series of the resistance exercise was performed with 50% of the maximum training load, the 2nd series with 75%, the 3rd with 90%, and the 4th with 100% of the set maximum load established. The protocol for adaptation and the physical effort test were executed as previously described for each training routine. The training is described in the Appendix A. All trained groups went through an adaptation phase 1 week before starting the exercise program.

In this protocol, for the aerobic exercise, the increase of length time for the first 3 weeks was lower compared to the aerobic training, and for the resistance exercise there was no addition of 30 g after 100% overload for the maximum load. 

The training protocol was calculated individually for each animal according to the tests carried out in all treated groups.

### 4.4. Euthanasia

The animals were euthanized at the end of the 16th week of the experiment. The groups fasted overnight (12 h) and were then anaesthetized with intraperitoneal xylazine (23 mg/kg) and ketamine (180 mg/kg). Blood was then extracted by heart puncture. The ATs (subcutaneous, retroperitoneal, and mesenteric) were collected immediately. The serum was then separated and the organs were transferred to labelled microtubes. The samples were then centrifuged at 19,293× *g*/15 min/4 °C and stored in a freezer at −80 °C.

### 4.5. Assessment of ELISA

FGF21 and irisin were determined by enzyme-linked immunosorbent assays (ELISA), using commercial kits from BT Lab (Jiaxing, Zhejiang, China), according to the manufacturer’s instructions.

### 4.6. Protein Extraction and Western Blotting

The AT samples (subcutaneous, retroperitoneal, and mesenteric) were homogenized with an extraction buffer of 1 M tris(hydroxymethyl)aminomethane hydrochloride (tris-HCl) at pH 7.4, 0.2 M ethylenediaminetetraacetic acid (EDTA), 10 mM sodium pyrophosphate, 100 mM sodium fluoride, 2 mM phenylmethylsulfonyl fluoride (PMSF), and 0.1 mg/mL of a protease “cocktail”, according to the amount in each sample. The samples were centrifuged at 19,293× *g*/40 min/4 °C, and the supernatant was removed. The protein content was analyzed using the spectrophotometer described above and the method by Bradford, using protein assay kits (Bio-Rad Laboratories, Hercules, CA, USA). Then, part of the supernatant was treated with the buffer described by Laemmli and heated in a dry bath at 97 °C for 5 min.

The samples underwent polyacrylamide gel electrophoresis using TGX™ FastCast™ kits (Bio-Rad Laboratories) in a Mini-PROTEAN device (Bio-Rad Laboratories) and in a running buffer (200 mM tris base, 1.52 M glycine, 0.4% sodium dodecyl sulfate, and ultrapure water). A Precision Plus marker (Bio-Rad Laboratories) of pre-established molecular weight was used on each gel. The transfer was performed electrically using a Transblot SD semi-dry transfer cell (Bio-Rad Laboratories). The membranes were pre-incubated for 2 h at 22 °C, to avoid non-specific protein binding, with a basal solution containing 1% BSA. The membranes were incubated overnight with specific antibodies diluted in basal solution containing 1% BSA. The following antibodies were used: AB23841 (Abcam plc, Cambridge, UK) for UCP1, #2430 (Cell Signaling Technology, Inc., Danvers, MA, USA) for PPARγ, Invitrogen PAS38021 (Thermo Fisher Scientific, Waltham, MA, USA) for PGC1α, SC-58667 (Santa Cruz Biotechnology, Inc., Dallas, TX, USA) for α-tubulin, and #8457S (Cell Signaling Technology, Inc.) for β-actin. Following this, the membranes were incubated with horseradish-peroxidase-conjugated secondary appropriate antibodies for 1 h. Then, ECL reagent (GE Healthcare Bio-Sciences, AB, Buckinghamshire, UK) was added and bands were detected by chemiluminescence in a documentation system (Alliance 4.7, UVItec, Cambridge, UK). The band intensities were determined by densitometry using Scion Image software (Scion Image version Alpha 4.0.3.2).

### 4.7. Statistical Analysis

The data were analyzed using the General Linear Model. The statistical model considered all the main effects and all possible interactions. The differences between individual means were later analyzed by completing pair comparisons (difference probability analysis). The Levene test was performed to assess the homogeneity between the groups, and either the Shapiro–Wilk or the Kolmogorov–Smirnov test was used to assess the normality of the variables. The Welch correction was applied for variables that did not meet the homogeneity prerequisite. The results of normal and homogeneous variables were expressed as mean ± standard error. A significance level of *p* ≤ 0.05 was considered. All statistical analyses were carried out using SPSS Statistics 21 (IBM Corporation, Armonk, NY, USA) and Prism 6.01 (GraphPad Software, San Diego, CA, USA) software. The individual data of each graphic are presented as Appendix A. 

## 5. Conclusions

The present study demonstrated that resistance and aerobic physical training were effective regarding the markers involved in the browning process of WAT in rats. After an obesity induction period, the combination of physical training with a normocaloric diet provided beneficial physiological changes in body mass parameters, adiposity, the relative mass of adipose depots, body fat, and the markers involved in the browning process. There are few studies in the literature that apply different training protocols to evaluate the browning process, and the present study is the first to compare resistance, aerobic, and combination training, as well as to analyze which types of training modified the concentration of those markers. These results showed that both resistance and aerobic training could promote physiological effects that are beneficial to health, by changing browning marker levels upregulating irisin, FGF21, PGC1α, PPARγ, and UCP1 in WAT; together, these may suggest an improvement in the thermogenic process in the adipose tissue, via the activation of the sympathetic nervous system, a potential control.

The findings of this study provide theoretical bases to help fight obesity and the metabolic disorders involved, demonstrating the impact of different protocols of physical training in this process. Healthy eating and physical training have potential beneficial effects on obesity control. These changes may modulate the expression of thermogenic biomarkers in white adipose tissue. Nevertheless, further studies are needed to investigate concurrent training under different protocols.

## Figures and Tables

**Figure 1 ijms-25-00275-f001:**
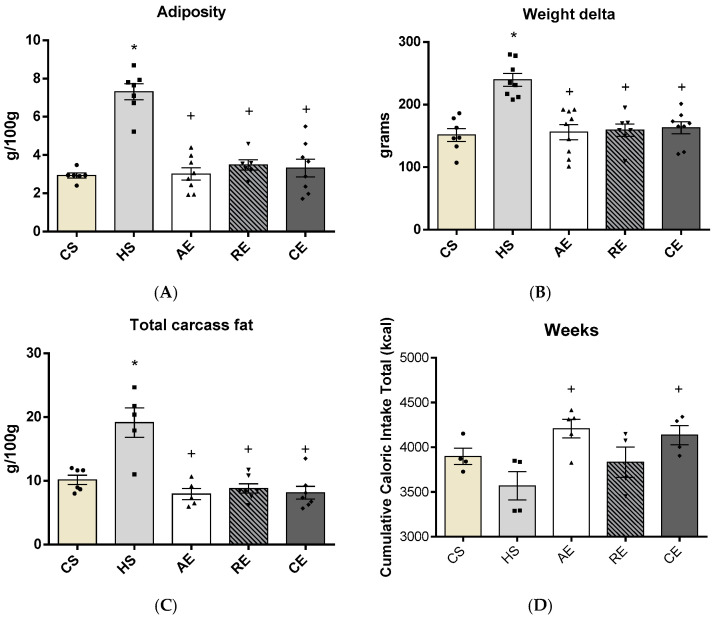
Body parameters after 16-week treatment and total food intake of CS, HS, AE, RE, and CE groups: (**A**) visceral adiposity (g/100 g), (**B**) delta weight (g), (**C**) total carcass fat (g/100 g body mass), and (**D**) cumulative caloric intake total (kcal). Samples varying from 4 to 9 animals; *: difference of CS; +: difference of HS group.

**Figure 2 ijms-25-00275-f002:**
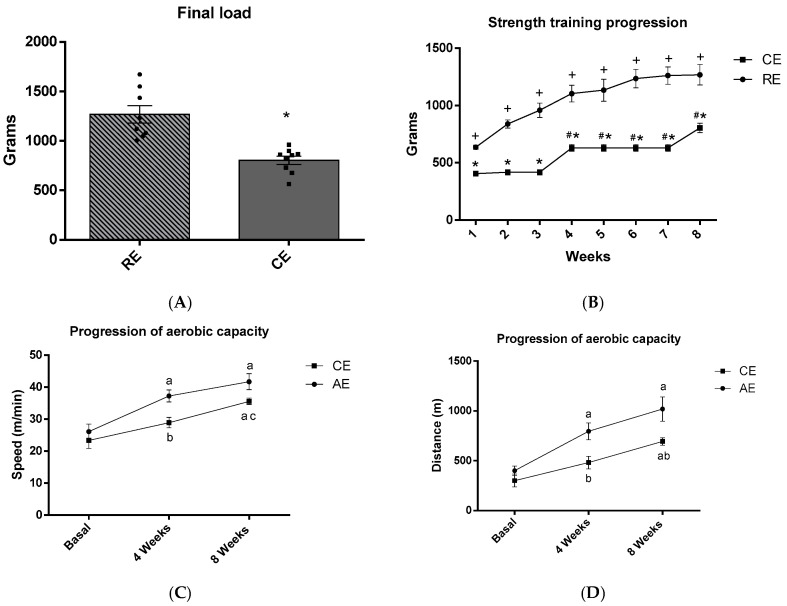
Load in grams of animals submitted to resistance or combined exercise and aerobic performance of animals submitted to aerobic or combined exercise: (**A**) final load (grams) of RE and CE groups, (**B**) strength training progression (grams) of RE and CE groups, (**C**) progression of aerobic capacity—speed (m/min), and (**D**) distance (m) of AE and CE groups during the 8 weeks of training. Samples varying from 8 to 9 animals; *: difference of RE group; #: difference of 1, 2, and 3 weeks in CE group; +: difference of 1 week; a: difference of difference of basal; b: difference of AE group; c: difference of 4 weeks.

**Figure 3 ijms-25-00275-f003:**
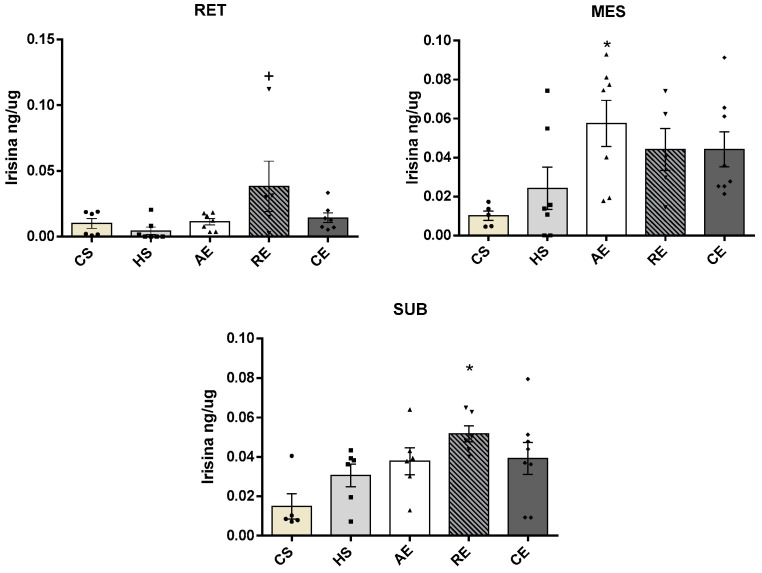
Adipose tissue irisin levels (ng/µg of total protein content) of CS, HS, AE, RE, and CE groups on adipose tissue: RET: retroperitoneal; MES: mesenteric; SUB: subcutaneous. Samples varying from 5 to 8 animals; *: significant difference of CS group; +: significant difference of HS group.

**Figure 4 ijms-25-00275-f004:**
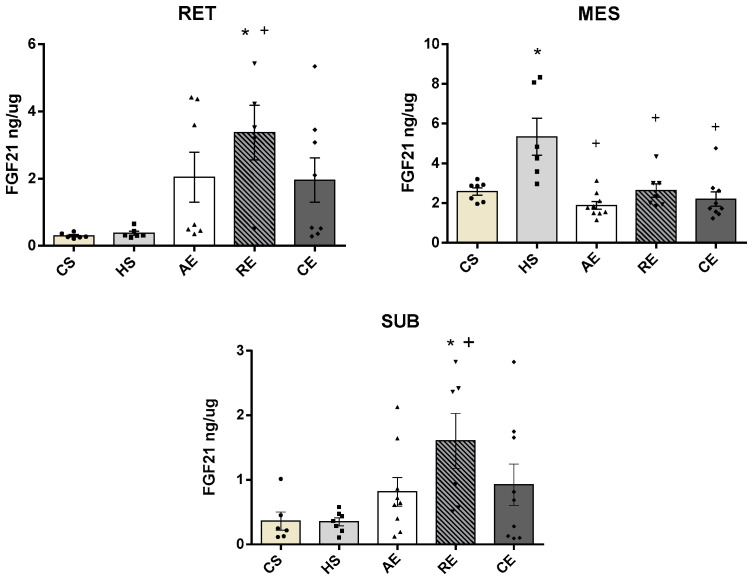
Adipose tissue FGF21 levels (ng/µg of total protein content) of CS, HS, AE, RE, and CE groups on adipose tissue: RET: retroperitoneal; MES: mesenteric; SUB: subcutaneous. Samples varying from 5 to 8 animals; *: significant difference of CS group; +: significant difference of HS group.

**Figure 5 ijms-25-00275-f005:**
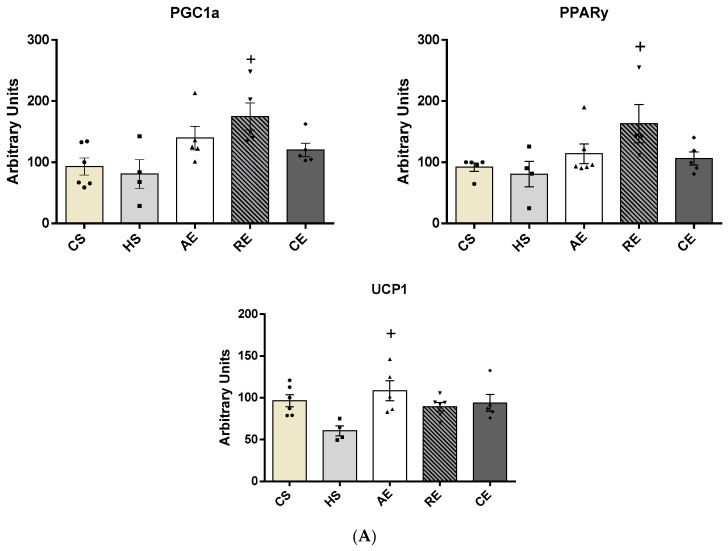
Mesenteric adipose tissue protein content by Western blotting from PGC1α (molecular weight: 90 KDa), PPARγ (molecular weight: 53/57 KDa), and UCP1 (molecular weight: 30 KDa) of CS, HS, AE, RE, and CE groups: (**A**) intensity of each band of the proteins analysed and (**B**) respective housekeeping protein (β-actin). Samples varying from 4 to 6 animals. +: significant difference of HS group.

**Figure 6 ijms-25-00275-f006:**
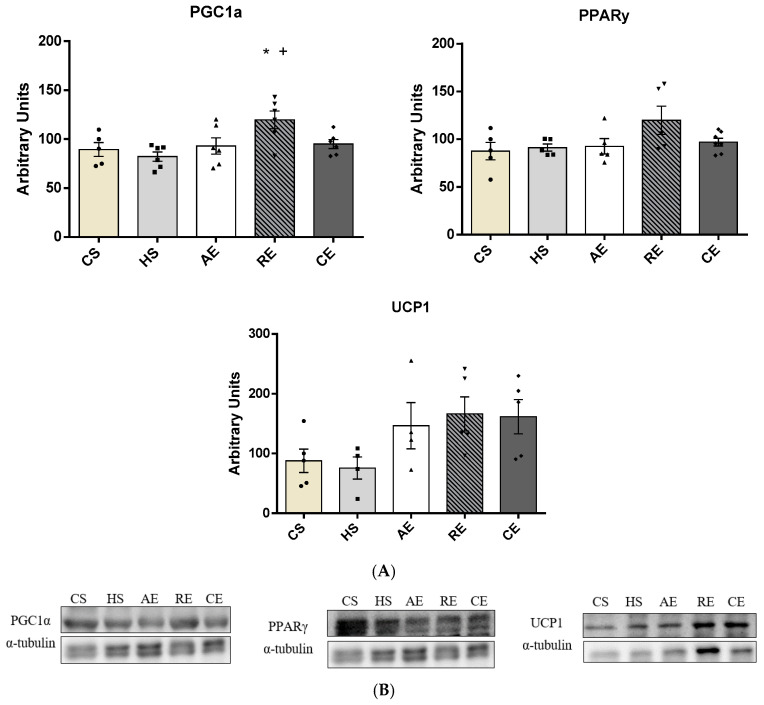
Subcutaneous adipose tissue protein content by Western blotting from PGC1α (molecular weight: 90 KDa), PPARγ (molecular weight: 53/57 KDa), and UCP1 (molecular weight: 30 KDa) of CS, HS, AE, RE, and CE groups: (**A**) intensity of each band of the proteins analysed and (**B**) respective housekeeping protein (β-actin). Samples varying from 5 to 7 animals. *: significant difference of CS group; +: significant difference of HS group.

## Data Availability

The data that support the findings of this study are available from the corresponding author upon reasonable request.

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
