# Peer review of "Resistance and Aerobic Training Were Effective in Activating Different Markers of the Browning Process in Obesity"

_ijms, 2023, doi:10.3390/ijms25010275_

Round 1
Reviewer 1 Report
Comments and Suggestions for Authors
The article by Lidia Passinho Paz Pontes addresses a potentially interesting topic: how exercise affects the energy balance via a thermogenic process process. This manuscript especially attempted to differentiate the effects of aerobic vs. resistant training using the rat model. However, overall, it is too descriptive and does not provide a causal relationship between exercise and the thermogenic process. There are some specific concerns.
1. It would be very helpful to have a diagram of experimental design. It was very difficult to follow how a combined exercise regime is different from individual training.
2. The point is how exercise affects thermogenic genes. Except for UCP1, other genes (PPARg, and PGC1) are not specifically related to brown or beigining adipogenesis. Hence, more brown/beige specific genes should have been examined.
3. An actual body weight needs to be reported, not just the delta body weight in Fig 1B.
4. This is related to point 2. Since very few genes are examined even though thermogenic processes require more complex pathways, it is hard to correlate changes in body weight vs gene expression.
5. It is very concerning why more than half of animals are dropped from analyses in each experiment/figure. There were n=16 for control groups and n=8 for exercise groups. However, Fig 1 shows the basic parameters such as adiposity, body weight, carcass fat, and yet only 4 -9 animals were used. All the other figures indicate 5-8 or 5-7 animals. For a bar graph, it would be better to show an individual data point.
Author Response
- It would be very helpful to have a diagram of experimental design. It was very difficult to follow how a combined exercise regime is different from individual training.
We appreciate the suggestion and include the experimental design diagram in the Supplementary material.
- The point is how exercise affects thermogenic genes. Except for UCP1, other genes (PPARg, and PGC1) are not specifically related to brown or beigining adipogenesis. Hence, more brown/beige specific genes should have been examined.
As we analysed some browning markers in the white adipose tissue, the reviewer is completely right to point out that many more components of thermogenesis pathways must be analysed to completely associate the body weight changes with this mechanism. How the browning process produces beige adipocytes in the white adipose tissue is still in discussion. As postulated by Sanchez-Gurmaches & Guertin, (2014) “Whether brite adipocytes form by trans-differentiation of existing white adipocytes or arise from a unique preadipocyte lineage is under debate…”. In 2017, Rui L. advanced the discussion and pointed out 3 mechanisms associated with beige adipocyte development. Under certain conditions, such as cold exposure, mature white adipocytes and beige adipocytes can be transdifferentiated in each other. Beige adipocytes may also exist in an inactive form and can be activated under certain conditions. The third mechanism is one mentioned by the reviewer, de novo beige adipogenesis. In this mechanism, the beige progenitor stimulation promotes the expansion and differentiation of mature beige adipocytes.
Bostrom et al. (2012) noted that trained muscles mimic fibronectin type III domain containing 5 (FNDC5) gene expression, which after cleavage is secreted into the bloodstream as irisin. Irisin is caused by physical training and activates profound changes in subcutaneous adipose tissue, stimulating darkening and UCP1 expression. It is worth highlighting the importance of this process in inducing the significant increase in total body energy expenditure and insulin resistance associated with obesity. Therefore, the action of irisin provides important benefits in response to exercise and muscular activity.
Important regulators of browning differentiation are PGC-1α and PPARγ, which play a fundamental role in mitochondrial biogenesis and act on NRF1 (nuclear interference factor 1) and TFAM (mitochondrial transcription factor A). These factors are regulators of respiratory chain genes and mitochondrial genome replication, respectively. PGC1α is stimulated by exposure to cold, and physical activity while fasting, and its function is adaptive thermogenesis through the nuclear receptor PPARγ (Cinti, 2018).
A circulating factor secreted after exercise and exposure to cold is FGF21, which has drawn attention due to its ability to regulate energy expenditure, metabolism of glucose, carbohydrates, lipids (Cuevas-Ramos et al. 2012), protection against oxidative stress and darkening of the WAT, increasing energy expenditure and weight loss (Fisher et al. 2012). This adipokine has mitogenic capacity, and presents important darkening properties, both directly in WAT and indirectly through increased sympathetic flow.
There are reports that this adipokine is secreted both in mice and in humans by adipocytes with an intermediate browning phenotype, thus reinforcing the browning characteristic of WAT. Furthermore, FGF21 induces glucose oxidation in many tissues, thereby promoting protection against obesity (Giralt et al. 2015; Ni et al. 2015).
Given this information, according to Cinti (2018), the lack of specifications in medical treatment is one of the causes that drive the scientific community to face the magnitude of the still unexplored molecular mechanisms in the physiology and pathophysiology of this important fatty complex, and an analysis of how these markers acting in response to changes in diet and physical exercise are of great relevance.
Bostrom, P.; Wu, J.; Jedrychowski, M.P.; Korde, A.; Ye, L.; Lo, J.C.; Rasbach, K.A.; Bostrom, E.A.; Choi, J.H.; Long, J.Z.; et al. A PGC1-alpha-dependent myokine that drives brown-fat-like development of white fat and thermogenesis. Nature. 2012; 481,463–468.
Cinti S. Adipose Organ Development and Remodeling. Compr Physiol. 2018; 14;8(4):1357-1431.
Cuevas-Ramos D, Almeda-Valdes P, Meza-Arana CE, Brito- Cordova G, Gomez-Perez FJ, Mehta R, Oseguera-Moguel J, Aguilar-Salinas CA. Exercise increases serum fibroblast growth factor 21 (FGF21) levels. PLoS One. 2012; 7:e38022.
Fisher FM, Kleiner S, Douris N, Fox EC, Mepani RJ, Verdeguer F, Wu J, Kharitonenkov A, Flier JS, Maratos-Flier E, Spiegelman BM. FGF21 regulates PGC-1alpha and browning of white adipose tissues in adaptive thermogenesis. Genes Dev. 2012; 26:271–81.
Giralt M, Cereijo R, Villarroya F. Adipokines and the endocrine role of adipose tissues. In: Switzerland SH, editor, Metabolic Control, Handbook of Experimental Pharmacology. Cham, Switzerland: Springer. 2015; 265-282.
Ni B, Farrar JS, Vaitkus JA, Celi FS. Metabolic effects of FGF-21: Thermoregulation and beyond. Front Endocrinol (Lausanne). 2015; 6: 148.
Rui L. Brown and Beige Adipose Tissues in Health and Disease. Compr Physiol. 2017 Sep 12;7(4):1281-1306. doi: 10.1002/cphy.c170001. PMID: 28915325; PMCID: PMC6192523.
Sanchez-Gurmaches J, Guertin DA. Adipocyte lineages: tracing back the origins of fat. Biochim Biophys Acta. 2014 Mar;1842(3):340-51. doi: 10.1016/j.bbadis.2013.05.027. Epub 2013 Jun 4. PMID: 23747579; PMCID: PMC3805734.
- An actual body weight needs to be reported, not just the delta body weight in Fig 1B.
Individual points have been included in the figure for 1B. We also included a table (Table 7) in the supplementary material that details the weighing of each group at the beginning, after eight weeks and after sixteen experimental weeks.
- This is related to point 2. Since very few genes are examined even though thermogenic processes require more complex pathways, it is hard to correlate changes in body weight vs gene expression.
As mentioned in the manuscript, this is one of few studies that compared different types of exercise protocols in the change of lifestyle to treat obesity. Our results confirmed that a change in the lifestyle, including a normolipidic diet and physical exercise is one of the best ways to decrease white adipose depot and body weight. Concerning the possible mechanism involved in the reduction of white adipose depot, we hypothesized that could be inducing the browning process, as some studies already showed that exercise can induce this process (this issue is already discussed in the manuscript). The most important point observed in this study was that different types of exercise protocols induce differently the white adipose browning markers. At this point, I would like to mention that, as already observed by reviewer 3, the gene expression analysis could be confirmed and improve the results from the protein quantification done. Unfortunately, we lost all the samples stored and doing a new batch will take too much longer.
As we confirm that the browning protein markers were modified by the different treatments, the next step will be to expand the study of pathways involved, by screening each mechanism of browning in the different adipose tissue depots.
Also, it is important to confirm that the decrease of adipose tissue depot is associated with increasing the thermogenesis and for this step, it is necessary to do the calorimetry analysis. Furthermore, the markers evaluated are considered relevant to evaluate the response to changes in diet and physical exercise, corroborating with Cinti (2018) on the importance of exploring the magnitude of the molecular mechanisms involved in the browning of white adipose tissue. Some of these aspects were included in the manuscript.
- It is very concerning why more than half of the animals are dropped from analyses in each experiment/figure. There were n=16 for control groups and n=8 for exercise groups. However, Fig 1 shows the basic parameters such as adiposity, body weight, carcass fat, and yet only 4 -9 animals were used. All the other figures indicate 5-8 or 5-7 animals. For a bar graph, it would be better to show an individual data point.
The sample size was divided into eight animals in each group. The sedentary groups had twice as many animals for the characterization of obesity. They were euthanized at the end of the experimental week as described in topic 2 and details without supplementary material (Chary 5). All analyses on browning markers were carried out considering the animals that reached the end of the experimental period. Throughout the experiment, we had some losses, and some were considered outliers, which is why they were not included.
Reviewer 2 Report
Comments and Suggestions for Authors
This is a well-designed and well-written manuscript to evaluate the effect of resistance, aerobic, and combination training on physiological changes and markers of browning of white adipose in rat models. The results showed that the combination of physical training with regular calorie intake is beneficial to both adipose tissue reduction and the browning process.
1. In the Introduction, the sentence “Recently, a pink AT ….. lactating rats” seems irrelevant and confusing when putting in this paragraph.
2. At the end of the 8-week obesity induction period, did they gain the same weight (non-significant) in groups HS, AE, RE, and CE, or did they have a similar weight when starting the training program at 8 weeks? What are the criteria for obesity in rats?
3. Is the total effort or exertion of the 8-week exercise equivalent, or at what percentage among aerobic, resistance, and combination groups? The information should be included in the Method section.
4. The authors should be cautious when making the conclusion that “combination training did not change browning marker levels” since the results did show decreased body parameters, and increased markers of browning in the combined training group but just not reach the significance, which might be due to short study period or limited rat numbers.
5. In supplementary Chart 6, the title column is confusing regarding the 8 weeks in AE and RE groups, and 16 weeks in CE group.
Author Response
- In the Introduction, the sentence “Recently, a pink AT ….. lactating rats” seems irrelevant and confusing when putting in this paragraph.
We appreciate the observation and removed the sentence.
- At the end of the 8-week obesity induction period, did they gain the same weight (non-significant) in groups HS, AE, RE, and CE, or did they have a similar weight when starting the training program at 8 weeks? What are the criteria for obesity in rats?
We included one table describing the body weight in 3 periods: at the beginning, after 8 weeks-treatment and after 16 weeks of treatment in the Supplementary material – Chart 7. The criteria for inducing obesity were described in item 2.1 Animals, Diet, and Experiment Groups: To assess the obesogenic effectiveness of the diet and validation of metabolic changes, some animals from groups CS (n = 8) and HS (n = 8) were euthanised after 8 weeks of obesity induction to check their biochemical and metabolic parameters during that period. Previous studies by our group have demonstrated the metabolic effects of the high-fat diet model (18, 19). The other animals from groups CS (n = 8) and HS (n = 8) were not euthanized until the end of the experiment to check the same parameters and compare them at the end. Body and biochemical parameters are described in Charts 5,6 and 8 of the supplementary material.
- Is the total effort or exertion of the 8-week exercise equivalent, or at what percentage among aerobic, resistance, and combination groups? The information should be included in the Method section.
We included in our methods that the protocol for each training was calculated individually for each animal according to the tests carried out. For the maximum effort test: the animals used the apparatus that allowed them repetitions to move from bottom to top on a ladder performed until failure. The overload was progressively increased with 50% of the maximum load in the first series, 75% in the second, 90% in the third and 100% in the fourth series, with 60-second intervals between series. In the following climbs, 30g was added in each attempt until failure. Regarding aerobic training: the animals were subjected to a physical exercise tolerance test with an initial speed of 5 m/min, increased every 3 minutes until the animal could no longer keep up with the running level. In the first week, the rats were trained at 30% of maximum speed (Vmax) and did not reach a tolerance test for 30 minutes. In the following 3 weeks, the duration was increased to 40, 50 and 60 minutes, respectively, with an intensity of 30% of Vmax. From the 5th week onwards, the duration was maintained, while the exercise intensity increased successively up to 40%, 45%, 50% and 55% of Vmax. The effort cannot be equivalent as they are different training sessions. The description of the tests is in section 2.3 of the method and details all types of training in the tables in the supplementary material.
- The authors should be cautious when concluding that “combination training did not change browning marker levels” since the results did show decreased body parameters, and increased markers of browning in the combined training group but just not reach the significance, which might be due to short study period or limited rat numbers.
We agree and the text was modified
- In supplementary Chart 6, the title column is confusing regarding the 8 weeks in the AE and RE groups and 16 weeks in the CE group.
We changed the description and content of the tables and added all information about body weight in chart 7 to be more understandable.
Reviewer 3 Report
Comments and Suggestions for Authors
In this study, it was observed that different exercise protocols have varying effects on the browning process in white adipose tissue. Resistance and aerobic training were found to efficiently activate distinct biomarkers associated with the browning process. The findings from this research suggest that exercise, particularly resistance and aerobic training, may hold therapeutic potential in combating obesity.
To improve the quality of this document, the following suggestions are provided:
In the abstract, authors should provide the full form of abbreviations when they are first introduced.
The authors mention, "Such stimuli that may increase thermogenesis include exposure to cold, diet, and response to physical exercise." It is important to note that thermogenesis can also be increased through drug administration, as discussed in the following article: https://doi.org/10.3390/cells10020403.
To confirm the results obtained through Western blot analysis, authors should consider performing RT-PCR on the RNA extracted from the tissues analyzing browning markers.
Authors could improve the structure of the document by separating the conclusion from the discussion paragraph into distinct sections.
Additionally, the authors might consider including a brief section describing the limitations of the study to provide a more comprehensive understanding of the research's scope.
Author Response
- In the abstract, authors should provide the full form of abbreviations when they are first introduced.
We include the full form of the abbreviations in the abstract.
- The authors mention, "Such stimuli that may increase thermogenesis include exposure to cold, diet, and response to physical exercise." It is important to note that thermogenesis can also be increased through drug administration, as discussed in the following article: https://doi.org/10.3390/cells10020403.
Thank you very much. The information was added to the manuscript.
- To confirm the results obtained through Western blot analysis, authors should consider performing RT-PCR on the RNA extracted from the tissues analysing browning markers.
We agree but unfortunately, we will not be able to carry out this analysis as we had lost stored tissue from the groups already treated. Furthermore, the treatment time is very long, and it would not be possible to carry out all treatments to obtain new tissue. Therefore, we will not carry out this analysis at this time.
- Authors could improve the structure of the document by separating the conclusion from the discussion paragraph into distinct sections.
We altered, as suggested.
- Additionally, the authors might consider including a brief section describing the limitations of the study to provide a more comprehensive understanding of the research's scope.
Thanks for the suggestion, we've included a section about limitations in the manuscript.
Round 2
Reviewer 1 Report
Comments and Suggestions for Authors
This is the revised version of the manuscript about the different exercise regimes and browning of adipose tissues. I don't think that the authors complied with any of this reviewer's criticism, even the simplest request, such as individual data points for the bar graph, which is standard practice these days. Also, I do not disagree that PPARg or PGC1g can participate in browning processes; however, there is also enough evidence that browning can be enhanced without any changes in these genes.
Author Response
"Please see the attachment."
Reviewer 3 Report
Comments and Suggestions for Authors
Authors responded to requests
Author Response
Thank you very much for your review. It certainly improved our manuscript.
Round 3
Reviewer 1 Report
Comments and Suggestions for Authors
i will sign off for its publication.